# A Printed Wearable Dual-Band Antenna for Wireless Power Transfer

**DOI:** 10.3390/s19071732

**Published:** 2019-04-11

**Authors:** Mohammad Haerinia, Sima Noghanian

**Affiliations:** 1School of Electrical Engineering and Computer Science, University of North Dakota, Grand Forks, ND 58202, USA; 2Phoenix Analysis and Design Technologies Inc., Tempe, AZ 85284, USA; sima_noghanian@ieee.org

**Keywords:** dual-band, wearable applications, Kapton

## Abstract

In this work, a dual-band printed planar antenna, operating at two ultra-high frequency bands (2.5 GHz/4.5 GHz), is proposed for wireless power transfer for wearable applications. The receiving antenna is printed on a Kapton polyimide-based flexible substrate, and the transmitting antenna is on FR-4 substrate. The receiver antenna occupies 2.1 ${\mathrm{cm}}^{2}$ area. Antennas were simulated using ANSYS HFSS software and the simulation results are compared with the measurement results.

## 1. Introduction

Wearable devices have been of interest due to an increase in their applications, such as the Internet of Things (IoT), biomedical sensors, and body area network. For these particular applications, a reliable and suitable power source is required, especially a method of power transfer that does not rely on batteries or wired power sources. Therefore, wireless power transfer (WPT) has been of interest for recharging wearable devices. 

Typically, the WPT systems are divided into two categories, far-field [1] and near-field [2,3] systems. A WPT system requires a transmitting unit connected to the main source of power and the transform the electrical power into an electromagnetic field. One or multiple receivers convert the electromagnetic field to electrical power. WPT technology is also used for energy transfer in wireless sensor networks (WSNs). In these applications, WSN is limited by the battery lifetime [2]. Flexible and textile implementations of WPT have been proposed in literature [4,5]. In [6], the authors designed a wireless power transfer system for a wearable and wireless neurotransmitter sensor recording system. They used a multi-antenna system as a power transmitter to resolve lateral and angular misalignments of the receiver antennas and claimed that it provided better effective coverage. In [7], the authors designed a wearable resonant WPT system for biomedical applications. This system achieved a power transfer efficiency (PTE) of 5.4% transferring at least 570 mW of power. Nonetheless, low power transfer efficiency and poor received power stability are the main drawbacks of WPT systems used in biomedical and wearable application [3]. A wearable textile antenna embroidered on fabric for wireless power transfer systems is presented in [8]. The authors used a planar spiral coil generated with the conductive thread on a cotton substrate. This system operates at 6.78 MHz, providing −5.51 dB transfer efficiency and 12.75 mW power transmission at a distance of 15 cm. The authors in [9] designed a single band antenna operating at 2.45 GHz intended for wearable and flexible applications. They took advantage of a thermal deposition technique to fabricate antenna structures on a Kapton Polyimide platform. In this case, thin-film deposition was obtained by evaporating a source material in a vacuum allowing vapor particles to travel to the substrate covered with a mask outlining the desired structure. In the work presented in [10], a low-cost inkjet printing method for antenna fabrication on a poly-ethylene terephthalate (PET) substrate is utilized. Their proposed co-planar waveguide (CPW) fed Z-shape antenna that was operating at 2.45 GHz. The authors measured radiation efficiency of 62% and the gain of 1.44 dBi at 2.45 GHz. An office inkjet printer was used to print silver nanoparticle ink on the PET substrate to fabricate the antenna prototypes. The authors in [11] fabricated a flexible dual-band dipole antenna operating at 900 MHz/2.44 GHz. They printed their antenna on Kapton through screen-printing technology using a polymer-silver conductor. This antenna is considered in its receiving mode and is connected to a rectifier. This radio frequency (RF) energy harvester was tested in a wireless power transfer scenario. In this dual-band configuration, the system provides 1 V DC voltage for a power density of 0.7 $\frac{mW}{m^{2}}$ at 900 MHz and 1.1 $\frac{mW}{m^{2}}$ at 2.44 GHz. The efficiency of WPT systems depends on different factors, such as the geometry of transmitting and receiving elements, misalignment, bending, and the distance between transceivers. Some of these were investigated in [12,13,14,15,16,17,18]. 

There are multiple challenges that should be addressed in using the wireless power transfer method as a source of power for wearable devices. One of the challenges is managing and distributing power between multiple wearable devices [19]. In [19], a WPT technique that distributes power from a single or a few sources between items of clothing is proposed. The authors also studied the power transfer between a pair of trousers and a shirt and provided three models of resonators attached to fabric on the surface of the clothing. In [20], the authors utilized an RF-based wireless power transfer method to transfer power to medical implanted devices, such as cardiac pacemakers. They designed a novel wideband numerical model (WBNM) for implantable antennas to enable RF-powered leadless pacing. The application of this model and the tissue simulating liquid (TSL) was demonstrated by the design, development, manufacture, and measurement of a novel metamaterial-based conformal antenna at 2.4 GHz. 

Despite the recent progress in implantable electronic devices, there is still a need for a reliable miniature power source. In [21] the authors studied the optimum frequency for power transfer to implanted devices. They concluded that the optimal frequency is above 1 GHz for small receive coil and typical transmit-receive separations. The author in [22] presented a wireless powering method that overcomes the challenge of miniaturization of the power source by inducing spatially focused and adaptive electromagnetic energy transport via propagating modes in tissue. This method has potential application in a new generation of micro-implants microelectromechanical sensors and logic units. 

In [23], WPT is used for wearable radio frequency identification (RFID). The authors of [23] illustrated that their proposed wireless transmission structure can operate RFID tags built into smartwatches and claimed that this technology can be adapted to various low-power chips to develop other smart wearable devices, such as smartphones, glasses, and bracelets. In [24], the authors designed a wearable motion sensor on a flexible substrate for mobile health applications. They proposed a compact wireless skin conductance sensor used to monitor the body’s emotional regulation process. This flexible substrate can conform to the shape of the user’s hand. 

The major part of any wireless flexible electronic system is the antenna. The antenna has a direct influence on the efficiency of the systems [4]. The size of the antenna is usually the limiting factor in achieving a reasonable power transfer efficiency. Therefore, antennas that can operate at different bands are desirable, since they can reduce the overall size of the system. In [25] we proposed a dual-band antenna and analyzed the bending effects on its performance. The antenna operates at two ultra-high frequency bands (1.6 GHz/3.6 GHz) [25]. The proposed antenna is for wireless power transfer for biomedical applications. This antenna is considered to be printed on a flexible substrate and be implanted inside human body tissue at 10 mm depth. The performance of the antenna under various bending conditions was studied. It was concluded that the proposed antenna respects safety standards to prevent dangerous effects in humans and the antenna’s performance remains stable under bending conditions. Following the work in [25], in this paper, we present a flexible antenna, printed on Kapton, with a total dimension of 15 mm × 14 mm × 0.17 mm, to be integrated with a WPT system. The design, simulation, fabrication, and measurements of the antenna radiation characteristics are presented. A comparison of different types of flexible antennas in the literature and the proposed work in this paper is shown in Table 1.

## 2. Design and Fabrication

It is important to choose a substrate material that is tested under bending conditions [26]. Bending, twisting, and rolling tests of the fabricated antenna prototypes presented in [27] show that Kapton is a robust material for designing wearable antenna. Kapton is prone to performance degradation because it resists substrate deformation. In this work, the flexible antenna was designed to be printed on Kapton. We chose Kapton because it has physical robustness, high flexibility, very high durability, high mechanical strength, distortion resistance to harsh environments, thus, enhancing the reliability of the antenna [4,28]. 

The printing material and equipment are described as follows [33]: 

*PCB Printer, Voltera V-One*: Voltera V-One shown in Figure 1 was used to fabricate the antenna. The Gerber file of the antenna design was first imported to V-one software. After calibrating ink and position, the device was set to dispense ink and solder paste onto the substrates.

*Conductive Ink*: The conductive ink from Voltera was used for printing transmitting antenna on the FR-4. After thermal curing the ink the antenna is ready for testing.

*Flexible Conductive Ink*: For the flexible substrate, the flexible conductive ink (from Voltera) that is compatible with Kapton (polyimide), polycarbonate, PET was used. This specific ink has a curing temperature of 140 °C for 10 min or 120 °C for 30 min.

The proposed microstrip antenna design is based on split-ring elements and can be used at two frequencies (2.5 GHz/4.5 GHz). The dimension of the transmitter (TX) and receiver (RX) antennas are 14 mm × 15 mm, occupying a small area.

In this design, a current-probe feeds the outer ring, and the inner ring is considered to provide frequency tuning. The design procedure is presented in [34]. To verify the design, antennas were simulated by ANSYS HFSS (High Frequency Structure Simulator) software [35]. Simulation results were compared with empirical ones. Figure 2 illustrates the proposed antenna. While the overall design of the TX and RX antennas is the same, the substrate materials and thicknesses are different. A rigid substrate (FR-4) with a thickness of 1.56 mm, relative permittivity of 3.66 and dielectric loss tangent of 0.004 was used for the TX. A flexible substrate (Kapton) with a thickness of 0.17 mm, relative permittivity of 3.4 and conductivity of 0.00524 $\frac{S}{m}$ was used for the RX. Since the substrate materials and thicknesses are different, that resonance frequencies of the TX and RX are not exactly the same. The antennas were simulated in the air and on the muscle tissue. The muscle tissue was assumed to have the relative permittivity of 49.54, and the conductivity of 4.0448 $\frac{S}{m}$ obtained from the Institute of Applied Physics (IFAC) database [36].

## 3. Experimental and Simulation Results

The location of the receiving antenna while it was placed on a phantom body model is presented in Figure 3a–c. The experimental setup is depicted in Figure 3d. As it is shown in Figure 4a the reflection coefficient for the TX antenna ($S_{11}$) is −19.50 dB at the first resonance frequency and −16.90 dB at the second resonance frequency. The reflection coefficient for the RX antenna ($S_{22}$) is −15 dB at the first resonance frequency and −21.90 dB at the second resonance frequency. These values are taken from the measurement in free-space. Similarly, Figure 4b presents the reflection coefficient for the TX antenna ($S_{11}$) is −17 dB at the first resonance frequency and −17.50 dB at the second one. The reflection coefficient for the RX antenna ($S_{22}$) is −23.60 dB at the first resonance frequency and −23.60 dB at the second one, while it is placed on the muscle tissue. $S_{21}$ in Figure 4a,b show the measured and simulated transmission coefficients for free-space and on body phantom cases, respectively. 

It is expected to see a slight difference in resonance frequencies of the TX and RX, due to the differences in the substrate’s materials. A slight shift in the measured resonant frequencies compared to simulated ones were due to fabrication discrepancies. Scattering parameters of these antennas at different resonance frequencies are shown in Table 2. In this paper, $S_{11}$ represents the reflection coefficient for the transmitting antenna (TX), $S_{22}$ provides the reflection coefficient for the receiving antenna (RX), and $S_{21}$ is the transmission coefficient. 

The simulated radiation patterns describe how the antenna radiates/receives energy into space. The antenna patterns are generally shown as plots in polar coordinates so the viewers have the ability to easily visualize how the antenna radiates in all directions. The ratio of the power gain in a given direction to the power gain of a reference antenna in the same direction defines the gain of the antenna [37]. The maximum gain of the TX at 2.5 GHz and 4.5 GHz is −5.34 dBi and −4.49 dBi, respectively, as shown in Figure 5. The radiation pattern was measured while the TX antenna is printed on an FR-4 substrate and was compared to simulation ones, as shown in Figure 6. The measured and simulated antenna specifications are listed in Table 3. 

Please note that in this case antennas are not necessarily in the far-field of each other. The antenna efficiency is not relevant in these applications. Instead, transmission efficiency is a good measure of the antenna’s performance. We studied the effects of distance on the proposed dual-band antennas’ transmission efficiency. To calculate transmission efficiency, we used (1).
(1)$$\eta ={\left|S_{21}\right|}^{2}\;\times 100\;\%$$
where *η* is transmission efficiency and S_21_ is the transmission coefficient. The distance was changed from 5 mm to 50 mm. The transmission efficiency variation was 3.94% for the first resonance frequency and 2.29% for the second one, as shown in Figure 7. It is concluded that the antenna performance is stable and does not change significantly due to this change of distance.

The maximum permissible exposure (MPE) in uncontrolled environments for electromagnetic field strengths is evaluated by the value of specific absorption rate (SAR). Based on IEEE Std. C95.1, SAR should be below 0.08 $\frac{W}{kg}$ as averaged over the whole body and the maximum SAR value should be below 1.6 $\frac{W}{kg}$ as averaged over any 1 g of the tissue. However, the maximum SAR value should be below 4 $\frac{W}{kg}$ as averaged over any 10 g of the tissue for the hands, feet, ankles, and wrists [38]. As it is indicated in Figure 8, the maximum absorption rate is 0.21 $\frac{W}{kg}$ at 2.5 GHz and 0.57 $\frac{W}{kg}$ at 4.5 GHz. The proposed antenna is within the safety standards. 

## 4. Conclusions

It is important to study flexible small antennas for the use of microwave power transfer (MPT) in wearable applications. In this work, we proposed a dual-band antenna operating at ultra-high frequency antenna (2.5 GHz/4.5 GHz) for MPT applications. The antennas were based on split ring resonator. The shape of the antenna provides multiple dimensions that can be optimized for the desired frequency. We proposed a small antenna with the area of 14 mm × 15 mm. The proposed design can be modified for other frequency bands. The proposed antenna provides near omni-directional radiation pattern that is desirable for wearable WPT. The transmission efficiency does not vary significantly due to change of distance. The SAR values were also examined. In future, the effect of bending and crumpling on the power efficiency will be investigated. The design will be optimized for various bending conditions.

## Figures and Tables

**Figure 1 sensors-19-01732-f001:**
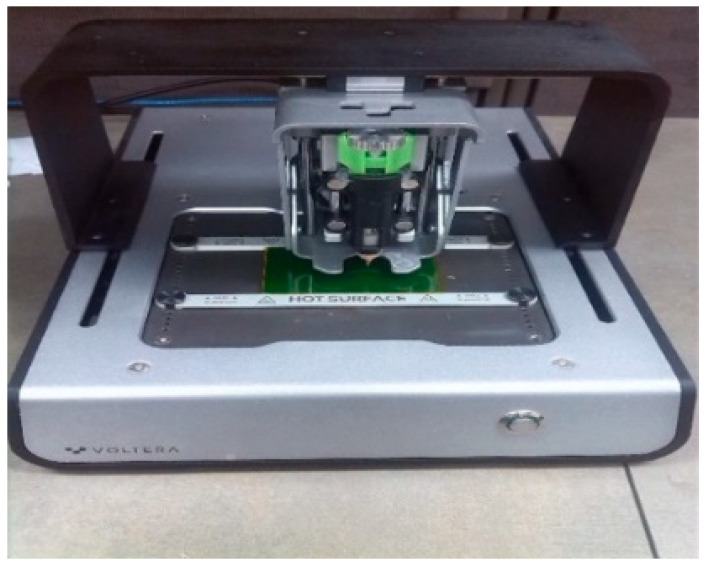
V-one from Voltera was used for fabrication.

**Figure 2 sensors-19-01732-f002:**
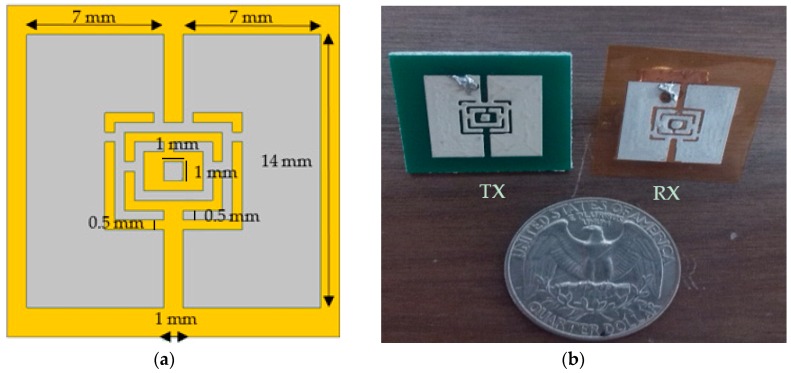
Transmitter (TX) and receiver (RX) antennas (**a**) antenna structure parameters, (**b**) TX and RX fabricated prototypes.

**Figure 3 sensors-19-01732-f003:**
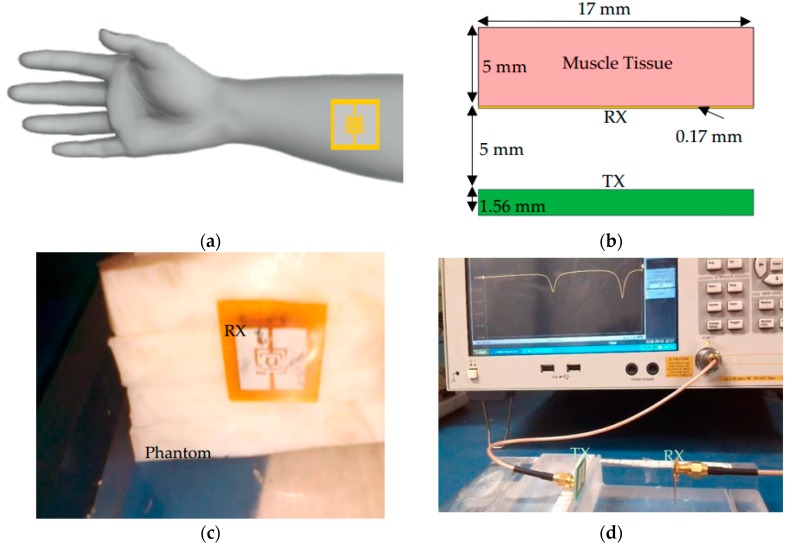
Proposed antennas (**a**) wearable antenna on hand, (**b**) antenna structure, (**c**) flexible antenna on phantom body model, and (**d**) experimental setup.

**Figure 4 sensors-19-01732-f004:**
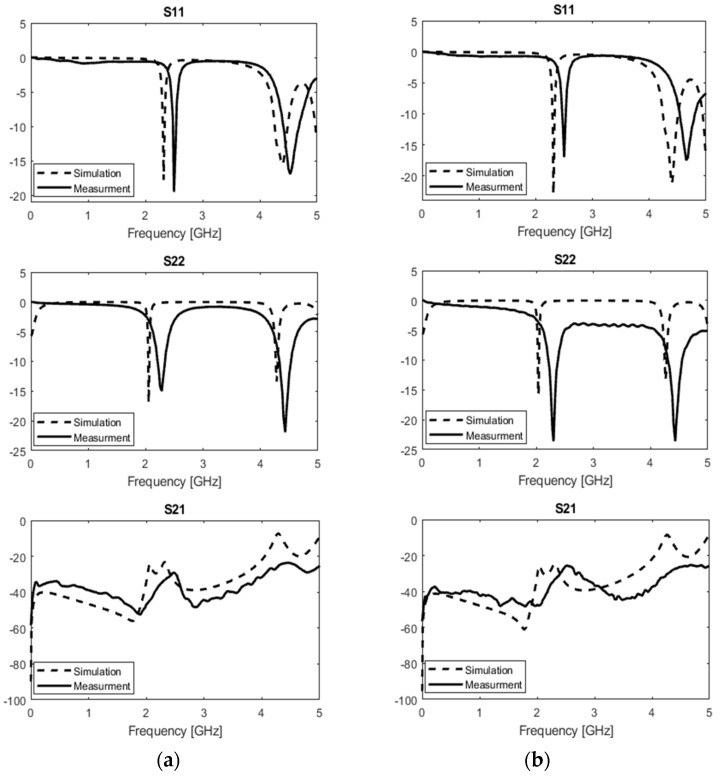
Scattering parameters $S_{11}$, $S_{22}$, and $S_{21}$ with 5 mm gap between the RX and TX, the RX is placed on (**a**) in free space, (**b**) on phantom.

**Figure 5 sensors-19-01732-f005:**
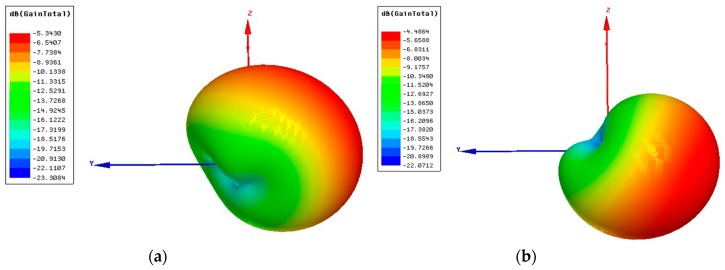
Total gain of the TX antenna (dBi) (**a**) at 2.5 GHz, (**b**) at 4.5 GHz.

**Figure 6 sensors-19-01732-f006:**
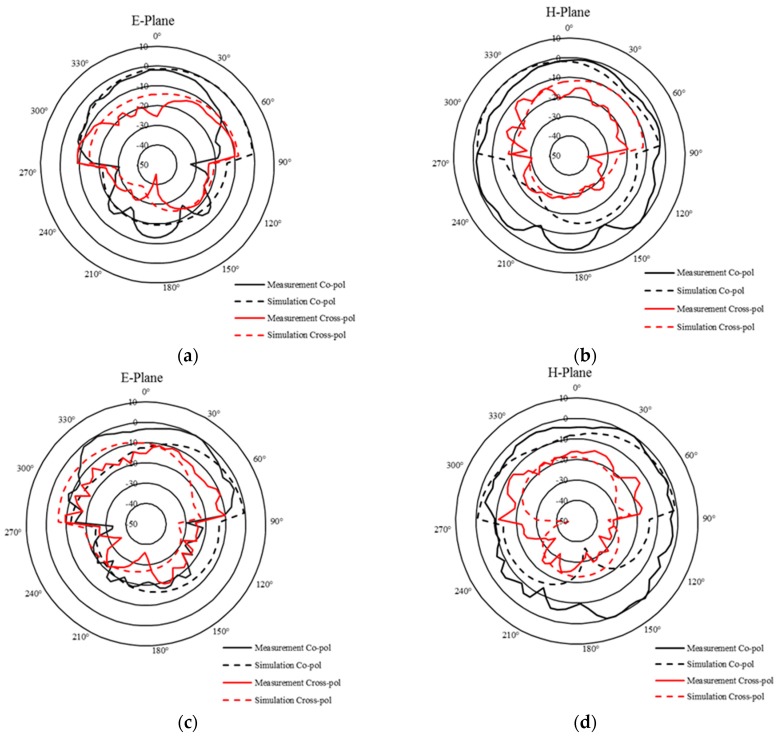
Radiation patterns of the TX antenna (dB) (**a**) E-plan at 2.5 GHz, (**b**) H-plan at 2.5 GHz, (**c**) E-plan at 4.5 GHz, (**d**) H-plan at 4.5 GHz.

**Figure 7 sensors-19-01732-f007:**
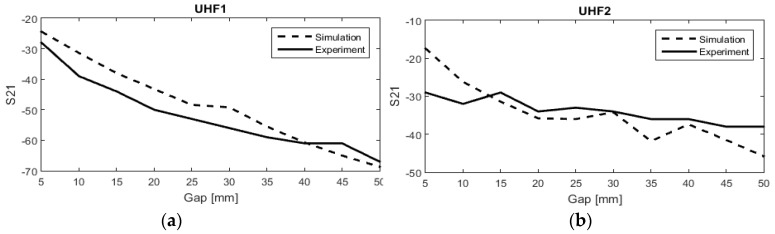
Simulated and measured transmission coefficient ($S_{21}$) versus distance between the transmitter (TX) and receiver (RX) antennas (**a**) at 2.5 GHz, (**b**) at 4.5 GHz.

**Figure 8 sensors-19-01732-f008:**
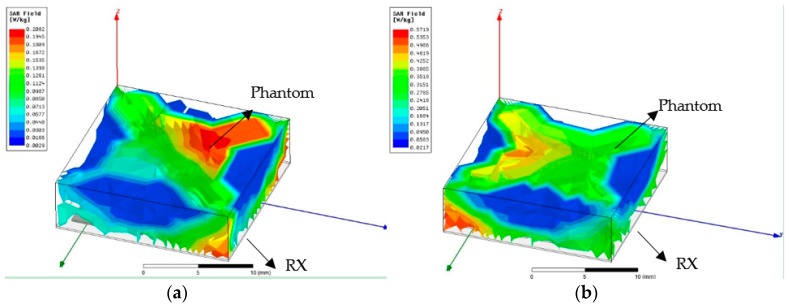
SAR distributions in Phantom (W/kg) (**a**) at 2.5 GHz, and (**b**) at 4.5 GHz.

**Table 1 sensors-19-01732-t001:** Comparison of different types of flexible antennas.

Antenna Parameter	Proposed Flexible Dual-Band Antenna	Poly-Imide Based Single Band Antenna [27]	Poly-Imide-Based Dual Band Antenna [27]	Textile Antenna [29]	Paper-Based Antenna [30]	Fluidic Antenna [31]	Flexible Bow-Tie Antenna [32]
Size (${\mathrm{mm}}^{2}$)	15 × 14	26.5 × 25	35 × 25	180 × 150	46 × 35	54 × 10	39 × 25
Thickness (mm)	0.17	0.05	0.05	4	0.25	1	0.13
Band	Dual	Single	Dual	Dual	Single	Single	Single
Frequency (GHz)	2.5/4.5	2.4	2.5/5.2	2.2/3	2.4	1.85	7.6
Substrate	Poly-imide	Poly-imide	Poly-imide	Felt fabric	Paper	PDMS	PEN film
Relative Permittivity ($\varepsilon_{r}$)	3.4	3.4	3.4	1.5	3.4	2.67	3.2
Deformability	Low	Low	Low	High	High	High	Low
Thermal Stability	High	High	High	Low	Low	Low	High
Fabrication Complexity	Simple/Printed	Simple/Printed	Simple/Printed	Complex/Non-Printed	Simple/Printed	Complex/Non-Printed	Simple/Printed

**Table 2 sensors-19-01732-t002:** Resonance frequencies (GHz) (M: Measurement, S: Simulation).

	Air (First Resonance)		Air (Second Resonance)		Phantom (First Resonance)		Phantom (Second Resonance)	
Parameter	M	S	M	S	M	S	M	S
$S_{11}$	2.50	2.32	4.53	4.39	2.50	2.31	4.65	4.44
$S_{22}$	2.28	2.05	4.43	4.28	2.30	2.03	4.43	4.26
$S_{21}$	2.48	2.32	4.45	4.28	2.51	2.31	4.68	4.26

**Table 3 sensors-19-01732-t003:** Antenna specifications (M: Measurement, S: Simulation).

First Resonance						Second Resonance		
Parameter	E-Plane M	E-Plane S	H-Plane M	H-Plane S	E-Plane M	E-Plane S	H-Plane M	H-Plane S
Directivity (dBi)	4.57	3.98	2.94	3.59	4.45	5.21	3.61	4.19
Gain (dBi)	−6.45	−5.34	−6.16	−5.34	−5.04	−4.49	−4.76	−4.49
Beam Width (°)	34	35	24	20	48	55	38	30

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
