# Peer review of "A Printed Wearable Dual-Band Antenna for Wireless Power Transfer"

_sensors, 2019, doi:10.3390/s19071732_

Round 1
Reviewer 1 Report
In this paper, authors design a dual-band printed planar antenna, operating at ultra-high frequency (2.5GHz /4.5GHz) . For wearable power transfer, it is an interesting topic. On the whole, this design is satisfied with some minor revision requirement.
What's the main motivation for the dual-band printed planar antenna, I can not find the enough background. Compared with previous design, what's the main differences between the design in this paper and the previous ones.
For Eq.1, authors should add the meaning of each parameter, in order the give more clear expression.
In this paper, there is no section for literature review, although some content in Introduction. As a regular research paper, why not give a detailed list for previous researches in this field. Maybe you can refer the content in “Wearable Vision Assistance System Based on Binocular Sensors for Visually Impaired Users” .
For conclusion, there is no application direction and future work.
Author Response
Authors would like to thank the editor, associate editor and the reviewers for their feedback. The manuscript was revised to address the questions and comments. The response to each of the comments is provided as follows.
In this paper, authors design a dual-band printed planar antenna, operating at ultra-high frequency (2.5GHz /4.5GHz). For wearable power transfer, it is an interesting topic. On the whole, this design is satisfied with some minor revision requirement.
Thank you for the comment.
What's the main motivation for the dual-band printed planar antenna, I cannot find enough background. Compared with the previous design, what're the main differences between the design in this paper and the previous ones?
Thank you for your comments. There are multiple designs of planar antennas. The use of antennas for microwave power transfer is somehow limited due to a low efficiency of this method, however, for applications which the receiver cannot be in the close vicinity of the transmitter, induction method is not efficient. We use dual-band frequency so it is possible to transfer power at both frequencies or use one frequency band for data communication. The use of dual-band reduces the overall size of the system and the number of needed antennas. This was the motivation behind the design of multi-band antenna. We have provided a table of comparison with other previously published systems in Table I, and compared them in terms of size, frequency band, substrate material, the relative permittivity
and other characteristics.
For Eq.1, authors should add the meaning of each parameter, in order the give more clear expression.
Your comment is appreciated. The explanation has been added to Eq.1.
In this paper, there is no section for literature review, although some content in Introduction. As a regular research paper, why not give a detailed list for previous researches in this field. Maybe you can refer the content in “Wearable Vision Assistance System Based on Binocular Sensors for
Visually Impaired Users”.
Thank you for your comments. The introduction is revised and now includes a lot more references. These references are added:
1. Garnica, J.; Chinga, R. A.; and Lin, J.; Wireless Power Transmission: From Far Field to Near Field, Proceedings of the IEEE, 2013, 101, 6, pp. 1321-1331. doi: 10.1109/JPROC.2013.2251411
2. Jawad, A.M.; Nordin, R.; Gharghan, S.K.; Jawad, H.M; Ismail, M. Opportunities and Challenges for Near-Field Wireless Power Transfer: A Review. Energies. 2017, 10, pp. 1022–1049.
6. Nguyen, C.M.; Kota, P.K.; Nguyen, M.Q.; Dubey, S.; Rao, S.; Mays, J.; Chiao, J.C. Wireless power transfer for autonomous wearable neurotransmitter sensors. Sensors (Switzerland) 2015, 15, pp. 24553–24572.
20. Asif, S.M.; Iftikhar, A.; Braaten, B.D.; Ewert, D.L.; Maile, K. A Wide-Band Tissue Numerical Model for Deeply Implantable Antennas for RF-Powered Leadless Pacemakers. IEEE Access 2019, 7, page 1.
21. Mekaladevi, V.; M, N.D.; Jayakumar, M. Design of SIW Cavity-Backed Antenna with Dual Dumbbell slot and Rectangular slot for 2.45GHz. 2018, pp. 1–7.
22. Patlolla, B.; Yeh, A.J.; Beygui, R.E.; Poon, A.S.Y.; Tanabe, Y.; Neofytou, E.; Kim, S.; Ho, J.S. Wireless power transfer to deep-tissue microimplants. Proc. Natl. Acad. Sci. 2014, 111, pp. 7974–7979.
23. Lin, D.B.; Wang, T.H.; Chen, F.J. Wireless power transfer via RFID technology for wearable device applications. 2015 IEEE MTT-S Int. Microw. Work. Ser. RF Wirel. Technol. Biomed. Healthc. Appl. IMWS-BIO 2015 - Proc. 2015, pp. 210–211.
24. Lam, L.K.; Szypula, A.J. Wearable emotion sensor on flexible substrate for mobile health applications. 2018 IEEE Sensors Appl. Symp. SAS 2018 - Proc. 2018, 2018–Janua, pp. 1–5.
25. Haerinia, M., Noghanian, S.: Study of Bending Effects on a Dual-Band Implantable Antenna'. USNC/URSI National Radio Science Meeting, Atlanta, Georgia, USA, 7-12 July 2019 (accepted).
For conclusion, there is no application direction and future work.
Thank you. The conclusion is revised and the future direction of the work is added.
The revised version of the paper now has better explanations, more references, and better resolution figures. We appreciate the reviewers’ comments which helped us improve the quality of the paper.

Reviewer 2 Report
Although the topic is interesting this paper suffers of serious flaws such as: poor image and graph quality, serious misunderstandings on S11. S22, S21. radiation patterns that do not look as expected, misunderstandings regarding the efficiency calculation Eq. (1), Fig. 8 is not the 'total gain of the TX antenna' in dBi but rather S21 in dB. beamwidths in Table 3 are completely unrealistic, there are no measured antenna gain, and the simulated gains of -5.34 and -4.49 dBi are not clear from Fig. 6.
Author Response
Authors would like to thank the editor, associate editor and the reviewers for their feedback. The manuscript was revised to address the questions and comments. The response to each of the comments is provided as follows.
Although the topic is interesting this paper suffers of serious flaws such as: poor image and graph quality, serious misunderstandings on S11. S22, S21. radiation patterns that do not look as expected, misunderstandings regarding the efficiency calculation Eq. (1), Fig. 8 is not the 'total gain of the TX antenna' in dBi but rather S21 in dB. beamwidths in Table 3 are completely unrealistic, there are no measured antenna gain, and the simulated gains of -5.34 and -4.49 dBi are not clear from Fig. 6.
Thank you for the comments. Please see the response below:
· Figures 5 and 8 (old Figures 6 and 9) are redrawn to make it more readable. We can provide image files with high resolutions.
· The following sentence was added to clarify the definition of S11, S22, and S21:
In this paper, S11 represents the reflection coefficient for transmitting antenna (TX), S22 provides the reflection coefficient for the receiving antenna (RX), and S21 is the transmission coefficient.
· It is not clear what the reviewer means by the radiation patterns do not look as expected. The simulation and measurement patterns are compared in Figure 6. The patterns generally follow the simulations.
· The explanation has been added to make Eq.1 clearer.
· The beam widths in Table. 3 were entered by mistake. The Table is corrected. Thank you for the note.
The revised version of the paper now has better explanations, more references, and better resolution figures. We appreciate the reviewers’ comments which helped us improve the quality of the paper.

Reviewer 3 Report
My comments:
The paper is intersting but can be improved in some aspects:
1) The introduction is very short and does not contain sufficient background about the state of art.
2) The sections 2 and 3 can be well described. They look like a summary.
3) The results can be well described, writing more text to explain the several figures.
4) The conclusion should be rewritten. Too short.
In summary, the paper should be well described, considering all sections.
Author Response
Authors would like to thank the editor, associate editor and the reviewers for their feedback. The manuscript was revised to address the questions and comments. The response to each of the comments is provided as follows.
The paper is interesting but can be improved in some aspects:
The introduction is very short and does not contain sufficient background about the state of art.
Thank you for your comments. The introduction is revised and now includes a lot more references. These references are added
1. Garnica, J.; Chinga, R. A.; and Lin, J.; Wireless Power Transmission: From Far Field to Near Field, Proceedings of the IEEE, 2013, 101, 6, pp. 1321-1331. doi: 10.1109/JPROC.2013.2251411
2. Jawad, A.M.; Nordin, R.; Gharghan, S.K.; Jawad, H.M; Ismail, M. Opportunities and Challenges for Near-Field Wireless Power Transfer: A Review. Energies. 2017, 10, pp. 1022–1049.
6. Nguyen, C.M.; Kota, P.K.; Nguyen, M.Q.; Dubey, S.; Rao, S.; Mays, J.; Chiao, J.C. Wireless power transfer for autonomous wearable neurotransmitter sensors. Sensors (Switzerland) 2015, 15, pp. 24553–24572.
20. Asif, S.M.; Iftikhar, A.; Braaten, B.D.; Ewert, D.L.; Maile, K. A Wide-Band Tissue Numerical Model for Deeply Implantable Antennas for RF-Powered Leadless Pacemakers. IEEE Access 2019, 7, page 1.
21. Mekaladevi, V.; M, N.D.; Jayakumar, M. Design of SIW Cavity-Backed Antenna with Dual Dumbbell slot and Rectangular slot for 2.45GHz. 2018, pp. 1–7.
22. Patlolla, B.; Yeh, A.J.; Beygui, R.E.; Poon, A.S.Y.; Tanabe, Y.; Neofytou, E.; Kim, S.; Ho, J.S. Wireless power transfer to deep-tissue microimplants. Proc. Natl. Acad. Sci. 2014, 111, pp. 7974–7979.
23. Lin, D.B.; Wang, T.H.; Chen, F.J. Wireless power transfer via RFID technology for wearable device applications. 2015 IEEE MTT-S Int. Microw. Work. Ser. RF Wirel. Technol. Biomed. Healthc. Appl. IMWS-BIO 2015 - Proc. 2015, pp. 210–211.
24. Lam, L.K.; Szypula, A.J. Wearable emotion sensor on flexible substrate for mobile health applications. 2018 IEEE Sensors Appl. Symp. SAS 2018 - Proc. 2018, 2018–Janua, pp. 1–5.
25. Haerinia, M., Noghanian, S.: Study of Bending Effects on a Dual-Band Implantable Antenna'. USNC/URSI National Radio Science Meeting, Atlanta, Georgia, USA, 7-12 July 2019 (accepted).
The sections 2 and 3 can be well described. They look like a summary.
Thank you for the comment. These sections are now removed and combined with the “Design and Fabrication” section.
The results can be well described, writing more text to explain the several figures.
The results are explained under “Experimental and Simulation Results”.
The conclusion should be rewritten. Too short.
Thank you. The conclusion is revised and the future direction of the work is added.
In summary, the paper should be well described, considering all sections.
Thank you. The revised version of the paper now has better explanations, more references, and better resolution figures. We appreciate the reviewers’ comments which helped us improve the quality of the paper.

Round 2
Reviewer 2 Report
The paper can now be accepted
Reviewer 3 Report
After the revisions, in my opinion, the paper is ready to be published.